REVIEW-SYMPOSIUM

# Oxyntomodulin physiology and its therapeutic development in obesity and associated complications

Martin T. W. Kueh[1,2], Ming Chuen Chong[3], Alexander D. Miras[4] and Carel W. le Roux[5]

[1] *UCD School of Medicine and Medical Science, UCD Conway Institute of Biomolecular and Biomedical Research, University College Dublin, Dublin, Ireland*
[2] *Royal College of Surgeons in Ireland and University College Dublin Malaysia Campus, Malaysia*
[3] *Royal College of Surgeons of Ireland, Dublin, Ireland*
[4] *School of Medicine, Ulster University, UK*
[5] *Diabetes Complications Research Centre, University College Dublin, Ireland*

The peer review history is available in the Supporting Information section of this article (https://doi.org/10.1113/JP287407#support-information-section).

**Abstract figure legend** Physiological influences of OXM and its emerging insights from evidence on bariatric surgery effects.
CCK, cholecystokinin; FGF21, fibroblast growth factor 21; GLUT4, glucose transporter type 4; IRS1, insulin receptor substrate-1; NNMT, nicotinamide *N*-methyltransferase; OXM, oxyntomodulin

**Professor Carel le Roux** graduated from medical school in Pretoria South Africa, followed by specialized training in metabolic medicine at St Bartholomew's Hospitals and the Hammersmith Hospitals, and obtained his PhD at Imperial College London. He later accepted his appointment as the Chair in Experimental Pathology University College Dublin (UCD). Currently, he holds the position as the Director of the Metabolic Medicine Group affiliated with the UCD Diabetes Complications Research Centre and leads the Innovative Medicine Initiative SOPHIA project. He has been the recipient of the following awards: President of Ireland Young Researcher Award, Irish Research Council Researcher of the Year Award, Irish Research Council Laurate Award, Clinician Scientist Award from the National Institute Health Research in the UK, and Wellcome Trust Clinical Research Fellowship. **Martin T. W. Kueh** is currently in his final year in the pursuit of a medical degree from the National University of Ireland. He completed his MSc in Medical Science at University College Dublin Diabetes Complications Research Centre.

The Journal of Physiology

**Abstract** Incretins, such as glucagon-like peptide-1 (GLP1) and glucose-dependent insulinotropic polypeptide (GIP), have advanced the treatment landscape of obesity to a new pinnacle. As opposed to singular incretin effects, oxyntomodulin (OXM) activates glucagon receptors (GCGR) and glucagon-like peptide-1 receptors (GLP1R), demonstrating a more dynamic range of effects that are more likely to align with evolving 'health gains' goals in obesity care. Here, we will review the molecular insights from their inception to recent developments and challenges. This review will discuss the physiological actions of OXM, primarily appetite regulation, energy expenditure, and glucose homeostasis. Finally, we will shed light on the development of OXM-based therapies for obesity and associated complications, and outline important considerations for more translational efforts.

(Received 29 July 2024; accepted after revision 8 October 2024; first published online 30 October 2024)

**Corresponding author** Carel W. le Roux: Diabetes Complications Research Centre, University College Dublin, Ireland, Belfield, Dublin 4, D04 V1W8, Ireland. Email: carel.leroux@ucd.ie

## Introduction

In conjunction with the relentless growth of cardio-metabolic diseases, obesity is now a worldwide epidemic (Lingvay et al., 2024). Bariatric surgery remains the most effective obesity treatment, largely mediated through rewiring hormonal and neural regulation (Lingvay et al., 2024). In the last decade, pharmacotherapy has been made available for the treatment by harnessing these key hormonal alterations (Melson et al., 2024). With the advent of incretin-based therapies, the treatment landscape of obesity reaches new benchmarks of 15%–25% over 70–90 weeks (Holst, 2024). In recent years, an oxyntomodulin (OXM)-based approach has stepped into the spotlight.(Melson et al., 2024)

In 1948, the identification of glucagon-like substances in the intestinal mucosa ushered in the discovery of OXM – a term derived from its role in influencing acid-secreting oxyntic glands (Bataille et al., 1982; Sutherland & de Duve, 1948). Due to its complex pharmacokinetic behaviours and the pharmacodynamic interplay between its agonism at glucagon-like peptide-1 (GLP1R) and glucagon (GCGR) receptors, its clinical advancement has lagged behind that of glucagon-like peptide-1 receptor agonist (GLP1RA) (Holst et al., 2018; Zhihong et al., 2023). Over the years, the paradigm of obesity care has shifted to increasing emphasis on health gains (Lingvay et al., 2024). The dual agonism of OXM provides a unique advantage of harmonising with this goal by exerting coordinated actions across multiple organs.

In this review, we discuss its molecular development and challenges in guiding therapeutic translation. Physiological influences of OXM will be addressed with a focus on appetite regulation, energy expenditure and glucose homeostasis (Abstract Figure). Finally, we discuss how the physiological understanding of OXM lays the groundwork for new treatments for obesity and obesity-related complications, as well as practical considerations as we move forward.

## Molecular insights into OXM

**Development.** OXM was first isolated and characterized in 1982. It has a low abundance ($\sim$0.5 mg OXM per ton) in porcine jejunum-ileum (Bataille et al., 1982), with an approximate measure of potency of roughly 20% for hepatic GCGR binding and 10% for adenylate cyclase stimulation (Bataille et al., 1982). OXM can represent about 40% of the enteroglucagons, with the highest concentrations seen in the ileum (Kervran et al., 1987). It also undergoes enzymatic processing in the jejunum-ileum and retains similar potency to full-length OXM after correcting for its higher metabolic clearance rate (Caries-Bonnet et al., 1991). OXM is less potent than glucagon in the liver and adipose tissues due to its C-terminal octapeptide (Caries-Bonnet et al., 1991; Schjoldager et al., 1989). However, it shows unique efficacy in gastric mucosal inhibition through a distinct receptor (Caries-Bonnet et al., 1991). Its secretion happens during digestion, peaking after meals and inhibiting histamine-stimulated gastric acid in a meal-dependent manner (Caries-Bonnet et al., 1991; Schjoldager et al., 1989). It also follows a diurnal pattern, with the highest concentration reached at night (Le Quellec et al., 1992; Schjoldager et al., 1989). OXM's metabolic clearance rate in humans was measured at 5.2 ml/kg/min, with a half-life of 12 min, slightly slower than glucagon (Schjoldager et al., 1988; Schjoldager et al., 1989). Two of its first recognised gastrointestinal regulatory effects are gastric acid secretion inhibition, which can be reduced by up to 76%, and a subtle incretin effect shown by a rise in insulin and C-peptide levels (Schjoldager et al., 1988).

**Structural processing.** Years of exploring OXM (Fig. 1) and its link to a larger precursor protein (i.e. proglucagon) have indicated that OXM is a proglucagon-derived 37-amino acid hormone (Holst et al., 2018; Zhihong et al., 2023). Proglucagon yields eight bioactive peptides,

primarily in the intestinal L-cells and the pancreatic alpha cells. In the intestinal L-cells, prohormone convertase 1/3 (PC1/3) processes proglucagon into glicentin, GLP1, GLP2 and OXM, and intervening peptides like IP-1 and IP-2 (Holst et al., 2018; Zhihong et al., 2023). OXM contains the full glucagon sequence (amino acids 33–61) plus an additional 8-amino acid extension at the C-terminus (position 62–69), giving it unique dual-agonism properties (Holst et al., 2018; Zhihong et al., 2023). OXM and glicentin share a highly similar sequence, with OXM containing the full glucagon sequence (position 33–61) and COOH-terminal octapeptide present within glicentin, and overlapping secretion patterns (Holst et al., 2018; Zhihong et al., 2023). It is now still unclear whether OXM is directly cleaved from proglucagon or arises due to glicentin instability. While PC1/3 is expressed in the nucleus of the solitary tract, its role in OXM synthesis outside the intestine remains uncertain (Holst et al., 2018; Zhihong et al., 2023). In contrast, the processing of proglucagon in the pancreas is catalysed by prohormone convertase 2 (PC2), leading to glucagon production. Little to no significant OXM or GLP1 was detected, except for pathological or adaptive conditions like pancreatic surgery or pregnancy (Holst et al., 2018; Zhihong et al., 2023).

**Challenges.** The molecular structure of OXM presents several challenges. First, due to its overlapping sequence and cross-reactivity with other peptides, as well as its inherently low traceable abundance levels, numerous commercially available ELISA kits have exhibited significant inaccuracies (Bak et al., 2014). Mass spectrometry validation is a more precise quantification method, using high-resolution accurate mass detection like orbitrap mass spectrometry, without relying on MS/MS fragmentation, to achieve ideal sub-picogram sensitivity (Cox et al., 2016).

Another challenge lies in measuring the potency of OXM on its receptors. One study has shown that OXM activated glucagon and GLP1R with half-maximal effective concentration ($EC_{50}$) values of 1.7 nM and 2.9 nM, respectively, compared to 107 pM and 72 pM for their native ligands (Holst et al., 2018). It showed no affinity for the glucose-dependent insulinotropic polypeptide (GIP) receptor (Holst et al., 2018). Although OXM has similar potency at respective receptors, its endogenous plasma levels are far below these potency thresholds (Holst et al., 2018). This suggests it acts rather locally, perhaps in a paracrine or autocrine manner, or through receptor sensitisation, resulting in distinct signalling pathways. Another study showed that isoquinoline compounds 14170 and 14175 enhanced OXM-mediated cAMP accumulation at the GCGR, demonstrating positive cooperativity, unlike glucagon, which showed negative modulation (Yuliantie et al., 2024). At the GLP1R, OXM presented with distinct $\beta$-arrestin 1 and 2 recruitment patterns, involving more receptor residues compared to GLP1 and exendin-4, and its internalization primarily relied on G protein-coupled receptor kinase 5/6 pathways (Lei et al., 2024; McNeill et al., 2024). This potent engagement led to biased activation of key metabolic regulators, such as AMP-activated protein kinase and protein kinase A, allowing unique hormonal reprogramming (Lei et al., 2024; McNeill et al., 2024).

The short half-life of OXM is caused by its susceptibility to degradation by DPP-4 and neutral endopeptidases (Holst et al., 2018; Zhihong et al., 2023). As a result, an early OXM analogue was developed in 2010 by attaching polyethylene glycol chains to the OXM peptide to improve its pharmacokinetic profiles (Kerr et al., 2010). Other subsequent modification approaches involve amino acid substitution (e.g. substituting serine at position 2 with $\alpha$-aminoisobutyric acid (Aib) or D-serine), fatty acid derivatization, and albumin fusion (Hope et al., 2021).

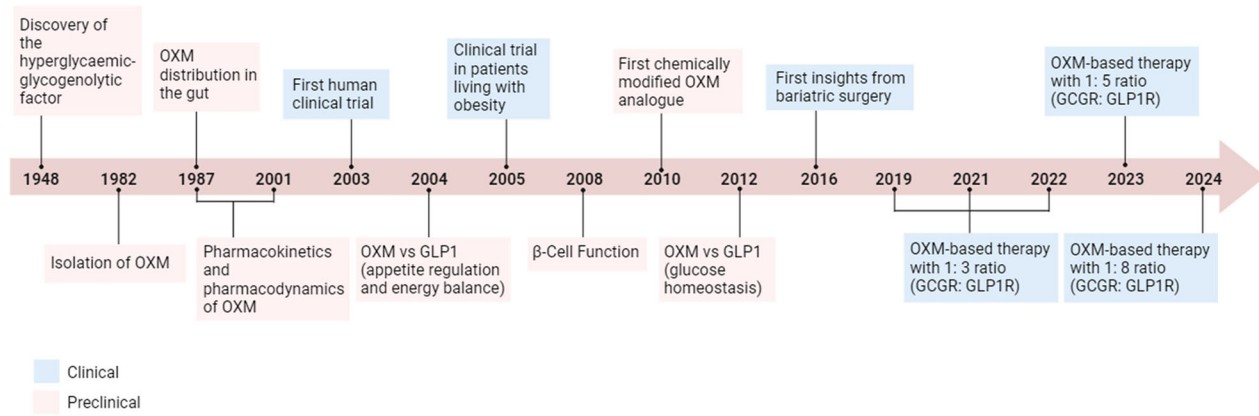

**Figure 1. Timeline showing key milestones in OXM discovery to therapeutic development.**
GCGR, glucagon receptors; GLP1, glucagon-like peptide-1; OXM, oxyntomodulin.

## Mechanisms of action of OXM

**Appetite regulation.** Early discoveries identified OXM as a major appetite regulator, as seen in its action in gastric acid secretion inhibition, delayed gastric emptying, reduced gastroduodenal motility, and decreased pancreatic enzyme secretion; an effect seen in its 19–37 fragments (cleavage at the Arg-Arg doublet), and in both fasting and postprandial states (Jarrousse et al., 1993; Schjoldager et al., 1989; Schjoldager et al., 1988). In a rat model study on intracerebroventricular and hypothalamic paraventricular nucleus administration of OXM, a comparable anorectic effect was noted despite OXM having a lower affinity for the GLP1R compared to GLP1 (half-maximal inhibitory concentration ($IC_{50}$): OXM – 8.2 nM; GLP1 – 0.16 nM) (Dakin et al., 2001). OXM's anorectic effect was blocked by the GLP1 antagonist exendin-(9–39), indicating that the effect is partly mediated through the GLP1. However, the fact that OXM still reduces food intake despite its lower affinity for the GLP1R suggests the possible involvement of central GCGR. While there is evidence of GCGR central processes, the exact localization of GCGR in the brain and bidirectional pathways of brain-pancreas-liver interaction are yet to be fully mapped (Wewer Albrechtsen et al., 2023).

The mounting delineation of specific hypothalamic nuclei localization found that, while OXM-induced satiety is mediated through highly conserved neural pathways in the arcuate nucleus (Baggio et al., 2004; Cline et al., 2008; Dakin et al., 2004; Halter et al., 2020), there are distinct central response patterns across species; for instance, activation of the dorsomedial nucleus in quail (Halter et al., 2020), downregulation of the lateral hypothalamus in broiler chicks (Cline et al., 2008), activation of the paraventricular nucleus in mice (Baggio et al., 2004), and activation of both the nucleus of the solitary tract and area postrema in mice (Baggio et al., 2004). More differential responses are evident in relation to the inconsistent anorexigenic responses to peripheral OXM administration, which was effective in rats (Dakin et al., 2004) and quail (Halter et al., 2020) but not in mice (Baggio et al., 2004) or chickens (Cline et al., 2008). In terms of gastric emptying, a reduction was seen in humans (Schjoldager et al., 1989), but not mice (Maida et al., 2008). These studies suggest that species-specific differences in OXM receptor signalling, metabolic pathways, and genetic predispositions exist, highlighting the need for human-centric research to draft a clearer roadmap for therapeutic breakthroughs.

Overall, the varied interspecies effects and ambiguous receptor engagement of OXM obscured a detailed scope of its appetite-related physiological processes. More recently, evidence from bariatric surgery may shed new light into its gut-brain axis modulation. Following gastric bypass, OXM levels increased >10-fold ($P = 0.001$; vs. pre-gastric bypass), and it was co-secreted and co-distributed with GLP1, which displayed a similar trend, increasing from $35\pm6$ pmol/g to $111\pm12$ pmol/g (Albrechtsen et al., 2016). This suggests that, in conditions driven by altered nutrient exposure and gastrointestinal adaptations, OXM may offer a more dramatic impact in appetite regulation (Albrechtsen et al., 2016). Assessment of hormones deriving from the posttranslational processing of proglucagon peptide found that OXM and glicentin (OXM precursor) had the largest increases in postprandial area under the curve (AUC) after bariatric surgery, ranging from 307.6% to 345.7% and 227.1% to 289.7%, respectively, from 3 months to 12 months (Perakakis et al., 2019). GLP1 showed modest AUC increases (23.6%–41.6%) over the same period, while glucagon showed minimal fluctuation (Perakakis et al., 2019). Comparing these hormones, OXM and glicentin were the most significant predictors of weight loss (Perakakis et al., 2019), suggesting that they were superior appetite regulators. Compared to other gut hormones, these two hormones together explain 60% of weight loss variation; 39% of the variation was accounted for by a shift in food preferences toward less energy-dense foods (Nielsen et al., 2020). The underlying mechanisms may involve reduced activation of the reward pathways (e.g. insula, putamen, and caudate), when viewing high-calorie food images (Perakakis et al., 2021). However, the implications of bariatric surgery to date should be treated cautiously since the findings were mainly based on associations; thus more mechanistic studies are required.

**Energy expenditure.** Earlier study identified that in chronic twice-daily intracerebroventricular OXM-treated rats, not only was food intake inhibited, but there was also an increase in energy expenditure (Dakin et al., 2002). The effect is conferred by GCGR, as indicated by studies showing that the effect of energy expenditure was independent of GLP1R but dependent on GCGR antagonism (Kosinski et al., 2012; Scott et al., 2018). Key energy-regulating biomarkers involved upon GCGR engagement include nicotinamide *N*-methyltransferase (NNMT), amino acids, and fibroblast growth factor 21 (FGF21) (Thomas et al., 2024; Zimmermann et al., 2022).

N-*Methyltransferase (NNMT)*. In lean mice treated with survodutide (BI456906), liver NNMT mRNA expression increased 15- to 17-fold relative to the vehicle control group (Thomas et al., 2024). Notably, higher NNMT expression correlates with greater GCGR potency (lower $EC_{50}$; $r = -0.47$; $P < 0.05$), reflecting its potential readout for GCGR agonism (Thomas et al., 2024). The effects of NNMT are not completely elucidated, and often depend on the context, tissue type (either adipocytes or hepatocytes)

and metabolic state. In adipose tissue, NNMT reduces energy expenditure by depleting $S$-adenosyl methionine (SAM) and nicotinamide adenine dinucleotide (NAD+), while its inhibition can increase energy expenditure through improved mitochondrial function and sirtuin 1 (SIRT1) signalling (Liang et al., 2023). Within the liver, controlled NNMT activity stabilizes SIRT1, promoting fat oxidation and increasing energy expenditure (Liang et al., 2023). This phenomenon is particularly expressed under nutrient-rich conditions, though prolonged overexpression may impair mitochondrial function and reduce energy expenditure (Liang et al., 2023). Aligned with the reduced liver triglycerides and liver marker (i.e. alanine aminotransferase) seen in survodutide-treated mice, this upregulation of NNMT could promote lipid mobilization and oxidation, as well as exert a hepatoprotective effect by involving detoxification and anti-oxidation (Liang et al., 2023; Thomas et al., 2024; Zimmermann et al., 2022). The effect appeared to be dose-dependent in survodutide-treated mice but was not seen in semaglutide-treated mice, suggesting that it is a determinant in energy intake and expenditure (Zimmermann et al., 2022). The mechanistic and causal impact of NNMT across the metabolic dysfunction-associated steatotic liver disease (MASLD) spectrum remains uncertain (Liang et al., 2023). Elucidating whether NNMT exerts protective effects in the early stages of MASLD or contributes to disease progression in later stages is an important question, as this distinction could influence therapeutic strategies.

*Amino acids.* In both acute and subchronic dosing studies, the effect of GCGR led to a significant dose-dependent reduction in plasma amino acids in survodutide-treated groups, a trend not observed in semaglutide-treated groups (Zimmermann et al., 2022). Notably, this effect was significantly pronounced in selected gluconeogenic amino acids such as alanine, citrulline, glutamine, ornithine, serine, threonine and tyrosine (Zimmermann et al., 2022). These findings suggest that GCGR activation specifically modulates hepatic catabolic processes, as evidenced by the down-regulation of dimethylarginine dimethylaminohydrolase 1 (DDAH1) and upregulation of enzymes like asparagine synthetase (ASNS), glutaminase 2 (GLS2), and glutamic-oxaloacetic transaminase 1 (GOT1), which correlated with reduced plasma serine and glutamine levels (Zimmermann et al., 2022). This promotes ureagenesis, facilitating the conversion of excess amino acids into urea, resulting in the establishment of a hypo-aminoacidaemic state that drives energy expenditure by transferring amino acids into energy-consuming pathways (e.g. tricarboxylic acid cycle) (Hope & Tan, 2023). Under excessive hepatic lipid accumulation, the GCGR gene is repressed (Eriksen et al., 2019).

The disruption in amino acid uptake and degradation contributes to the worsening of hepatic pathogenesis, with the impact more pronounced in other amino acids. Some examples include branched-chain amino acids, which impair glucose and lipid metabolism; aromatic amino acids (e.g. tryptophan, phenylalanine and tyrosine), which are associated with steatosis and inflammation progression; methionine, homocysteine, and arginine, which cause derailment in oxidative stress. Given their significance for MASLD, expanding understanding of these amino acids would be valuable (Gaggini et al., 2018; Tricò et al., 2021).

*Fibroblast growth factor 21 (FGF-21).* The upregulation of FGF21 with GCGR, with a strong potency in an *in vitro* model (lower $EC_{50}$; $r = -0.67$; $P < 0.01$) supports its role in increasing energy expenditure (Thomas et al., 2024). Survodutide at 20 nmol/kg elicited higher FGF-21 levels than semaglutide at 100 nmol/kg, signifying FGF21's potent engagement with GCGR (Zimmermann et al., 2022). FGF21, largely hepatic in origin, regulates energy metabolism through the $\beta$-klotho/FGFR1 receptor complex, which is selectively expressed in key metabolic tissues such as adipose tissue, liver and the central nervous system (Geng et al., 2020). Within adipocytes, FGF21 induces browning of white adipose tissue while promoting brown adipose tissue activity through the upregulation of uncoupling protein 1 (UCP1) and peroxisome proliferator-activated receptor gamma coactivator 1-alpha (PGC1$\alpha$) (Geng et al., 2020). Simultaneously, this upregulation facilitates glucose uptake and disposal by insulin-independent glucose transporter type 1 (GLUT1) in adipocytes, which explains glucagon's contribution to glucose homeostasis, albeit more downstream and somewhat indirect (Geng et al., 2020). The therapeutic effects on MASLD are probably explained by its equally profound, coordinated actions in reducing triglyceride accumulation via stimulating fatty acid oxidation, inhibiting lipogenesis, and improving insulin sensitivity through suppression of mechanistic target of rapamycin complex 1 (mTORC1) and activation of PGC1$\alpha$ (Geng et al., 2020). Evident in chronic glucagon administration, its effect may extend centrally in the hypo-thalamus, where it could amplify sympathetic outflow and potentiate thermogenesis (Hope & Tan, 2023; Nason et al., 2021). An emerging phenomenon – FGF21 resistance in obesity, due to diminished sensitivity of its coreceptor $\beta$-klotho, raises concern on the rewiring energy balance in dual MASLD-obesity burden.

**Glucose metabolism.** A few years after the discovery of OXM, Schjoldager et al. found increased plasma insulin and C-peptide levels following high-dose OXM administration (Schjoldager et al., 1988). The possibility of an incretin effect with insulinotropic properties has

broadened focuses on decoding the role of OXM in glucose regulation and meal-induced insulin secretion.

In C57Bl/6 mice with diet-induced insulin resistance, OXM improved glucose tolerance via increasing plasma insulin concentrations, improving glucose disposal, and suppressing glucose production under hyperinsulinaemic conditions (Parlevliet et al., 2008). OXM reduces $\beta$-cell apoptosis, preserves pancreatic function for sustained insulin secretion and induces the trans-differentiation of pancreatic $\alpha$-cells into $\beta$-cells, a process particularly relevant in both type 1 and type 2 diabetes where $\beta$-cell loss is prominent (Maida et al., 2008; Zhihong et al., 2023). OXM was believed to regulate glucose metabolism through a pathway similar to, but not dependent on GLP1 (Pocai, 2014). One study showed that GLP1R agonism (via OXMQ3E) was superior to OXM in modulating glucose metabolism without causing ketogenesis, confirming that GLP1 is a more dominant insulinotropic receptor (Du et al., 2012).

One critical consideration is whether glucagon co-activation will function synergistically or lead to hyper-glycaemia. Treatment with DualAG (protease-resistant dual GLP1R/GCGR agonist) in a mouse model showed improvement in glucose tolerance in diet-induced obese mice without causing hyperglycaemia, unlike typical GCGR activation; all while inducing weight loss, steatosis attenuation and lipid-lowering effects (Pocai et al., 2009). Interestingly, OXM still induced insulin secretion in GLP1R$-/-$ mice, suggesting that GCGR activation can contribute to the insulinotropic action of OXM, probably through direct stimulation of $\beta$-cells (Du et al., 2012). To ascertain the synergistic effects, a double-blinded study of 4-h continuous infusions among 15 healthy lean men was conducted to examine the effects of isolated or combined GLP1R and GCGR (Bagger et al., 2015). OXM, GLP1 and glucagon all reduced food intake to a comparable degree. Glucagon led to a slight increase in postprandial glucose levels; however, when administered with GLP1, it counterbalanced the glucose-raising effects (Bagger et al., 2015). The positive metabolic effects of OXM, characterized by increased insulin secretion and reduced glycaemic excursion, appeared comparable to liraglutide in individuals with obesity and type 2 diabetes (Pocai et al., 2009). Notably, OXM improved glucose homeostasis by augmenting glucose-dependent insulin secretion, independently of weight loss (Pocai et al., 2009).

Interestingly, improved glucose regulation with OXM analogues, particularly the acylated version ((dS2)Oxm[K-$\gamma$-glu-Pal]), may have downstream central effects, including improved hippocampal neuro-genesis, synaptogenesis, and reduced oxidative damage (Pathak et al., 2015). At the molecular level, the analogues upregulated the expression of insulin signalling-related genes in the hippocampus, such as glucose transporter type 4 (GLUT4) and insulin receptor substrate-1 (IRS1)

(Pathak et al., 2015). A randomized controlled trial within a prediabetic population found significantly lower levels of OXM in glucose responders, despite unchanged glucagon, GLP1, glicentin, and major proglucagon fragment levels, following ketone $\beta$-hydroxybutyrate supplementation (Liu et al., 2024). Reduction of OXM levels resulted from a compensatory response. However, uneven changes in glucagon and GLP1 suggest potential involvement of central neural circuits rather than just peripheral actions (Liu et al., 2024). Unlike peripheral GCGR activation, which raises blood glucose by increasing hepatic glucose production, central GCGR signalling may inhibit glucose production via vagal signalling (Liu et al., 2024). This implies that OXM, by means of central GCGR signalling, may offset the conventional hyperglycaemic glucagon effects, thereby promoting glucose homeostasis (Liu et al., 2024). The authors also pointed to a potential inverse relationship between cholecystokinin (CCK) and OXM. Low OXM increases CCK, enhancing exocrine pancreatic function and potentially improving glucose metabolism by improving digestion and nutrient absorption (Liu et al., 2024). This could lead to increased secretion of digestive enzymes, such as amylase and lipase, linked to improved glucose metabolism and improved $\beta$-cell function.

## Progress of OXM-based therapies

In the earliest appetite regulation study in humans, acute intravenous infusion of OXM in 13 healthy adults resulted in a significantly reduced energy intake ($-19\%$), suppressed fasting plasma ghrelin ($-44\%$), and decreased cumulative 12-h energy intake ($-11.3\%$) (Cohen et al., 2003). Among individuals who were overweight and obese ($n = 26$), subcutaneous OXM significantly decreased body weight ($-2.4\%$) and reduced energy intake by up to 35%, with changes in adipose hormones (i.e. lower leptin, higher adiponectin), after 4 weeks (Wynne et al., 2005). Over a longer duration of 10 weeks, OXM reduced energy intake ($-17.3\%$), and increased activity-related energy expenditure (26.2%) along with long-term efficacy in weight loss (Wynne et al., 2006). Collectively, these studies suggested the potential of OXM as a therapeutic candidate.

Many trials have been conducted since but have failed due to the difficulty in identifying a suitable glucagon-to-GLP1 ratio. A balanced synergistic effect between GLP1 and glucagon is needed for maximizing systemic benefits; while imbalance – whether through overexpression of glucagon or under expression of GLP1 – can precipitate pathophysiological consequences. These include hyperglycaemia, perturbed muscle catabolism, dysregulated amino acid metabolism, altered lipid profiles or cardiovascular effects such as increased heart rate and blood pressure (Holst et al., 2018; Zhihong et al., 2023). NNC9204-1177, made up of a ratio of 1:3 (GCGR:

GLP1R) was terminated due to its side effects, including elevated heart rate, reduced reticulocyte count, and heightened inflammatory markers (Friedrichsen et al., 2023). The dose-dependent impact of these adverse effects is probably a reflection of the physiological limits of receptor activation. Trials of SAR425899 (GCGR:GLP1R ratio 1:5) were also discontinued due to intolerable gastro-intestinal side effects, possibly because the ratio was still unbalanced, resulting in disproportionate metabolic shifts (Eriksson et al., 2020).

Gastrointestinal side effects are a common occurrence in appetite regulation studies, with similar effects seen in a GLP1 mono agonist study (Holst, 2024). A systematic review and meta-analysis of 14 trials on cotadutide (MEDI0382) and mazdutide (IBI362 or LY3305677) showed that there are significantly higher odds of treatment-emergent adverse events (odds ratio (OR): 2.52; 95% confidence interval CI: 1.92–3.30) and vomiting (OR: 6.05; 95% CI: 3.52–10.40) (Deng et al., 2024). While weight loss was evident, a phase 2a trial of cotadutide found that the treatment arm was associated with a reduction in physical activity, probably due to adverse effects of nausea, lethargy and loss of appetite (Golubic et al., 2024). It has been proposed that, at least during the first 6 weeks, cotadutide can induce a 1.5–2.5 times greater weight loss than that observed with GLP1RA (Golubic et al., 2024). It is critical to address the adverse effects to ensure they do not inadvertently mediate the weight loss response.

Survodutide, with a GCGR:GLP1R ratio of 1:8, is now thought to provide a more appropriate benefit-to-risk ratio. A phase 2 trial of a once-weekly subcutaneous injection of survodutide randomized among 387 participants with obesity found a dose-dependent weight loss, with the highest dose (4.8 mg) resulted in a 14.9% weight reduction at week 46 (le Roux et al., 2024). With a view to treating MASLD by combining the extrahepatic effect of GLP1 with the direct hepatic effect of GCGR, another phase 2 study involving 293 individuals with fibrosis stages F1 to F3 and a mean BMI of 35.81 ± 6.41 kg/m² found a dose-dependent effect of survodutide on weight loss (highest dose (6.0 mg) group lost ∼8%–10%) but not on liver fat reduction (greatest reduction of ∼67% seen in 4.8 mg group) (Sanyal et al., 2024). This result suggests that a fine balance between the two receptors is important to improve hepatic lipolysis and fat oxidation without over-stimulating gluconeogenesis (Sanyal et al., 2024). We are unsure of survodutide's impact on earlier or later stages of MASLD given our unclear physiological understanding of NNMR on MASLD progression, and a long-term study is needed to determine if FGF21 resistance affects the obesity group. Should it have an influence, how to bypass resistance or enhance FGF21 sensitivity is the next question.

## Practical directions for OXM-based therapies

A range of new obesity pharmacotherapies mimicking natural hormones is emerging, sparking curiosity about their comparative effectiveness. A 16-week exploratory comparative analysis was promising, with survodutide significantly reduced HbA1c and body weight in a dose-dependent manner, with higher doses leading to greater body weight loss compared to semaglutide (up to −8.7% in survodutide ≥1.8 mg qw vs. −5.3% in semaglutide 1.0 mg qw) (Blüher et al., 2024). In survodutide trials (NCT04667377 and NCT04771273), (le Roux et al., 2024; Sanyal et al., 2024) gastrointestinal side effects were most evident (i.e. nausea 56%–66%, diarrhoea 22%–49% and vomiting 27%–41%) being consistently high across all doses, except for 0.6 mg dose. A 20% to 25% discontinuation rate (vs. 3% to 4% in the placebo group) was most seen in the rapid escalation phase, which could be reduced through more gradual titrations. A mild increase in pulse rate and pancreatic hyperenzymaemia were reported as adverse effects, which are probably due to the response of glucagon (le Roux et al., 2024; Sanyal et al., 2024). In developing this treatment, titrating up doses is expected to present a challenge similar to GLP1 mono agonist, but with a trickier physiology to tackle (Holst, 2024). At the current juncture, the development of clinical data shows promise with an optimal activity ratio of 1:8 between the two agonists. An ongoing 114-week phase 3 trial (NCT06077864) will affirm survodutide's long-term efficacy and safety, with composite cardio-vascular outcomes as the primary endpoint.

In contrast to existing weekly injection protocols, G3215, a real-time 14-day continuous subcutaneous infusion, is being trialled (Hope et al., 2024). This adaptive infusion approach allows for rapid titration, leading to a mean loss of 2.39 kg in 14 days with fewer gastrointestinal side effects (Hope et al., 2024). The evolving landscape of pharmaco-development has initiated a multidimensional optimization, with evolving innovation in dose and regimen strategy. Finally, OXM-based therapies promise alignment with the shift in models of care towards health gains. Analogues have demonstrated efficacy in preserving islet architecture and providing renal protection by mitigating fibrosis (Zhihong et al., 2023). OXM also strengthens bone and collagen, while exerting neuroprotective effects by reducing inflammation, improving cognitive function, and protecting against neurodegenerative diseases like Parkinson's and Alzheimer's (Zhihong et al., 2023).

## Conclusion

OXM will probably redefine the landscape of obesity and metabolic therapies, providing an unparalleled physio-logical approach by simultaneously targeting GLP1R and

GCGR. This dual agonism, beyond its effect on two pathways, provides a renewed opportunity to revisit the bidirectional, compensatory, and synergistic actions that govern appetite control via gut-brain interaction, hepatic-to-systemic energy expenditure, and glucose homeostasis via insulin-glucagon balance. Early barriers to therapeutic translation, such as its rapid renal clearance and vulnerability to enzymatic degradation, are being overcome by modified analogues that extend its half-life and bioactivity. The upcoming generation of OXM-based therapies will allow the identification of the optimal ratio of dual agonism for maximum benefit-to-risk ratio. It has the potential to have an efficacy comparable to bariatric surgery, offering a powerful, non-invasive option to patients with obesity and the complications of obesity.

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

## Additional information

### Competing interests

Carel W le Roux receives grants from the Irish Research Council, Science Foundation Ireland, Anabio, and the Health Research Board. He serves on advisory boards of Novo Nordisk, Herbalife, GI Dynamics, Eli Lilly, Johnson & Johnson, Glia, and Boehringer Ingelheim. He is a member of the Irish Society for Nutrition and Metabolism outside the area of work commented on here. He was the chief medical officer and director of the Medical Device Division of Keyron in 2011. Both of these are unremunerated positions. He was a previous investor in Keyron, which develops endoscopically implantable medical devices intended to mimic the surgical procedures of sleeve gastrectomy and gastric bypass. The product has only been tested in rodents and none of Keyron's products are currently licensed. They do not have any contracts with other companies to put their products into clinical practice. No patients have been included in any of Keyron's studies and they are not listed on the stock market. Professor le Roux was gifted stock holdings in September 2021 and divested all stock holdings in Keyron in September, 2021. He continues to provide scientific advice to Keyron for no remuneration. All other authors declare that they have no known competing financial interests or personal relationships that could have appeared to influence the work reported in this paper.

## Author contributions

Conceptualization: M.T.W.K., M.C.C., A.D.M., C.W.leR.; writing, original draft, M.T.W.K.; writing, review and editing, M.T.W.K., M.C.C., A.D.M., C.W.leR. All authors have approved the final version of the manuscript and agree to be accountable for all aspects of the work. All persons designated as authors qualify for authorship, and all those who qualify for authorship are listed.

## Funding

This research did not receive additional financial support from organizations beyond the authors' academic institutions.

## Keywords

energy expenditure, glucose metabolism, gut-brain axis, obesity, oxyntomodulin

## Supporting information

Additional supporting information can be found online in the Supporting Information section at the end of the HTML view of the article. Supporting information files available:

**Peer Review History**

