## [Peer Review History · The Journal of Physiology]

Oxyntomodulin physiology and its therapeutic development in obesity and its complications

Martin Tze Wah Kueh, Ming Chuen Chong, Alexander D Miras, and Carel Le Roux
DOI: 10.1113/JP287407

Corresponding author(s): Martin Tze Wah Kueh (martin.kueh@ucdconnect.ie)

Review Timeline:

Submission Date:	29-Jul-2024
Editorial Decision:	14-Aug-2024
Revision Received:	23-Sep-2024
Accepted:	08-Oct-2024

Senior Editor: Kim Barrett

Reviewing Editor: Kim Barrett

Transaction Report:

Dear Dr Le Roux,

Re: JP-SR-2024-287407 "Harvesting how the gut talks to the brain: Oxyntomodulin based treatments for obesity and its complications" by Martin Tze Wah Kueh, Ming Chuen Chong, Alexander D Miras, and Carel Le Roux

Thank you for submitting your manuscript to The Journal of Physiology. It has been assessed by a Reviewing Editor and by 2 expert referees and we are pleased to tell you that it is acceptable for publication following satisfactory revision.

ABSTRACT FIGURES: Authors may use The Journal's premium BioRender account to create/redraw their Abstract Figures (and any other suitable schematic figures). Information on how to access this account is here: <https://physoc.onlinelibrary.wiley.com/journal/14697793/biorender-access>.

REVISION CHECKLIST: Upload a full Response to Referees file. To create your 'Response to Referees' copy all the reports, including any comments from the Senior and Reviewing Editors, into a Microsoft Word, or similar, file and respond to each point, using font or background colour to distinguish comments and responses and upload as the required file type.

- 'Potential Cover Art' for consideration as the issue's cover image.
- Appropriate Supporting Information (Video, audio or data set: see https://jp.msubmit.net/cgi-bin/main.plex?form_type=display_requirements#supp).

We look forward to receiving your revised submission.

Yours sincerely,

REQUIRED ITEMS

- Please include an Abstract Figure file, as well as the Figure Legend text within the main article file. The Abstract Figure is a piece of artwork designed to give readers an immediate understanding of the Review Article and should summarise the main conclusions. If possible, the image should be easily 'readable' from left to right or top to bottom. It should show the physiological relevance of the Review so readers can assess the importance and content of the article. Abstract Figures should not merely recapitulate other figures in the Review. Please try to keep the diagram as simple as possible and without superfluous information that may distract from the main conclusion of the Review. Abstract Figures must be provided by authors no later than the revised manuscript stage and should be uploaded as a separate file during online submission labelled as File Type 'Abstract Figure'. Please ensure that you include the figure legend in the main article file. All Abstract Figures will be sent to a professional illustrator for redrawing and you may be asked to approve the redrawn figure before your paper is accepted.

- The reference list must be in alphabetical order, rather than numbered, to comply with our Journal format.

- Author profile(s) must be uploaded via the submission form. Authors should submit a short biography (no more than 100 words for one author or 150 words in total for two authors) and a portrait photograph of the two leading authors on the paper. These should be uploaded and clearly labelled together in a Word document with the revised version of the manuscript. Any standard image format for the photograph is acceptable, but the resolution should be at least 300 DPI and preferably more. A group photograph of all authors is also acceptable, providing the biography for the whole group does not exceed 150 words.

EDITOR COMMENTS

The reviewers agreed that this article addresses an important area. But they also have some meaningful suggestions for revision that I trust will be useful in revising the manuscript. The editor would particularly emphasize carefully distinguishing between physiological and pharmacological effects, and also the possibility to add some useful graphics, which tend to substantially increase the impact of topical reviews such as this.

Please also see 'Required Items' above.

REFEREE COMMENTS

Referee #1:

There are a number of issues in this short overview on oxytomodulin (OXM) that need to be addressed.

1. There are no page numbers.

2. There are no figures. As for any review article, figures are critical to summarize and transport the main message(s) put forward by the authors.

3. There are numerous previous review articles on the subject which provide good summaries on OXM biology and pharmacology, including PMID 23019069 (2012), 29412831 (2018), PMID 37917871 (2024)

4. The Introduction section is very brief and thus requires more detail about the molecular biology and molecular pharmacology of OXM. PMID 29412831 provides one summary "Oxyntomodulin is a product of the glucagon precursor, proglucagon, produced and released from the endocrine L-cells of the gut after enzymatic processing by the precursor prohormone convertase 1/3. It corresponds to the proglucagon sequence 33-69 and thus contains the entire glucagon sequence plus a C-terminal octapeptide, comprising in total 37 amino acids. As might have been expected, it has glucagon-like bioactivity, but also and more surprisingly also activates the receptor for GLP-1." The authors should also mention that OXM is one of 8 peptides proteolysed from the glucagon peptide (glicentin, glicentin-related polypeptide (GRPP), glucagon-like peptide 1 (GLP-1, incretin), glucagon-like peptide 1 (7-37), glucagon-like peptide 1 (7-36), glucagon-like peptide 2 (GLP-2) (PMID 10605628 and <https://www.uniprot.org/uniprotkb/P01275/entry>), and that proteolytic cleavage may be dependent of the cell type and organ, as in In contrast, in L-cells , proglucagon is cleaved by PC1/3 at Arg-Arg sites to yield glicentin, GRPP, oxyntomodulin (OXM), GLP-1, intervening peptide-2 (IP-2) and GLP-2 (PMID10499540, PMID 9074764). It would be helpful if the authors could summarize the molecular biology and molecular pharmacology in a figure.

5. "...a randomised trial found significantly lower levels of OXM in glucose responders without significant changes in the levels of GLP-1 and glucagon.⁶⁷ The actions of GLP1 and glucagon on their respective receptors were insufficient to explain the decrease in blood glucose." This conclusion is overinterpreting the findings not what the study showed, the aim of which was "To investigate changes in plasma levels of PGDP in glycaemic responders versus non-responders." in prediabetic individuals with samples being collected for 2.5 hours after consuming a ketone β -hydroxybutyrate (KE β HB)-supplemented beverage. The authors concluded that "Oxyntomodulin is involved in lowering plasma glucose and may play an important role in diabetes remission" which also is not supported by data, because changes in circulating levels are affected by production and clearance and say nothing about peptide activity.

6. Lines 190ff and lines 212-220: These sections require more detail on the molecular biology and pharmacology of FGF-21, particularly the comprehensive literature published by Aimin Xu's group on the role of FGF-21 needs to be discussed here

7. Lines 361 ff: "Conclusion: Emerging without a designated receptor and regarded as less potent to native ligands (i.e., GLP1 and glucagon). This statement contradicts the literature and the authors' own statements in the manuscript (Lines 60 ff: "60 Oxyntomodulin (OXM), produced by the endocrine cells in the gut, activates glucagon receptors (GCGR) and glucagon-like peptide-1 receptors (GLP1-R)." and Lines 75 ff: "Oxyntomodulin (OXM), a 37 amino-acid hormone derived from proglucagon splicing in intestinal endocrine cells, acts as an endogenous dual agonist on glucagon receptor (GCGR) and glucagon-like peptide-1 receptor (GLP1-R).^{1, 2} Its potency is lower than the native ligands, due to its C-terminal"). Also see comment #4 above

Referee #2:

. This review discusses the physiological mechanisms induced after administration of oxyntomodulin, a glucagon-containing proglucagon derived peptide from the gut which activates both glucagon and GLP-1 receptors. The current interest in the peptide is due to the recent development of glucagon/GLP-1 coagonists which to a varying extent activate both receptor and therefore resemble oxyntomodulin. The review discusses these mechanisms and the translational aspects.

Comments.

I am somewhat confused about the review. A lot of literature is cited but rather uncritically, species are mixed between each other, and it seems that the authors do not distinguish at all between the natural oxyntomodulin paper and its physiology and the more recently developed coagonists. Already in the summary and the introduction I miss a discussion of the physiological versus the pharmacological importance of oxyntomodulin. While its physiological role is likely to be limited because of low concentrations of the endogenous peptide and poor affinity for its two receptors, analogs with higher affinities that can be administered in much higher doses may have marked effects. The uninformed reader may think that

oxyntomodulin is an important endogenous hormone. It is probably not. It is fine to write a review on the coagonists and their pharmacological actions, but that has nothing to do with oxyntomodulin. Of course, if high doses of oxyntomodulin are administered there may be effects from more or less unphysiological activation of the GCGR and GLP-1R, but that is of limited interest.

In addition the review is poorly written - several sentences are incomprehensible and others misinterpret the references provided. Thus as a whole the review is often misleading and of little use for the interested reader.

Specific points

L. 94 : there is no evidence that oxyntomodulin actually binds to these regions of the brain

I. 107-113 the authors apparently suppose that peripheral oxyntomodulin actually enters the arcuate nucleus, but there is no evidence for that (whereas oxyntomodulin may certainly interact with the GLP-1 receptors of the circumventricular organs)

I. 114 "the interspecies expression of OXM in central ARC or peripheral total alimentary canal transit time reduction" I don't understand this sentence (and if I do, it is wrong)

I. 120-130: the effects of OXM on the clock are demonstrated with pharmacological doses and do not reflect physiological actions of the peptide

I. 132-143. It is a bit bold to ascribe the effects of bariatric surgery to the limited rises in low-affinity endogenous oxyntomodulin

I. 146: the authors should note that glucagon interacts with both the glucagon and the GLP-1 receptors

I. 151-162: this section is very confusing and poorly written. It is impossible to understand the point (except that oxyntomodulin should be measured with specific sandwich assays - and such assays are currently not available (in a reliable form))

I. 164. One of the paradoxes in the literature was that the energy increasing activity of oxyntomodulin was found to be due to increased physical activity. This of course is difficult to explain and highly unlikely

I. 183-4: "Low blood insulin levels lead to an energy insufficiency, causing the release of glucagon to increase energy expenditure processes. This is a concentration-substantiated process mediated through brown adipose tissue or the sympathetic nervous system" I don't know what was the intention with this sentence, it is (again) incomprehensible, and at any rate not consistent with current physiological knowledge

I. 186: there is no evidence that glucagon is taken up by the liver cells (it acts on them)

I. 182-193: do the authors suggest that glucagon activates energy metabolism in humans via an action on the brown adipose tissue??

I. 198: do the authors really believe that glucagon acts via the FCXR? Glucagon increases fatty acid oxidation in the liver and inhibits de novo lipogenesis and VLDL export, via interaction with the GCGR and the cAMP pathways. Those provide good explanations for the effects

I. 235 "As discussed earlier, GLP1-R activation is rather insulin-centric, leading to glycogenesis and acetyl-CoA conversion to fatty acid-mediated through protein kinase C and PPAR gamma signalling". Again this is impossible to understand or wrong

I. 247: there is no evidence that GLP-1 acts directly on the liver

I. 320: endogenous oxyntomodulin is not substrate for DPP-4

I. 330: it should be noted that the development of cotadutide is currently put on hold

I. 346: "Overall, we are now witnessing the OXM-led physiological reprogramming and how its interplay with obesity associated pathology in human trials." Another example of the problematic style of this review.

Overall, Regarding the clinical application: it is rather simple: what happens depends of the ratio between glucagon activity and GLP-1 activity of the dual agonist; if there is too much glucagon all the problems with glycemic control, amino acid metabolism, side effects, and cardiac effects set in, but there is a stronger effect on hepatic lipid metabolism.

END OF COMMENTS

Editor comments

The reviewers agreed that this article addresses an important area. But they also have some meaningful suggestions for revision that I trust will be useful in revising the manuscript. The editor would particularly emphasize carefully distinguishing between physiological and pharmacological effects, and also the possibility to add some useful graphics, which tend to substantially increase the impact of topical reviews such as this.

Dear Editor,

We wish to express our sincere appreciation for the opportunity afforded to us to revise our work. In response to the comments, we have made careful, and significant revisions to our manuscript, now titled 'Oxyntomodulin physiology and its therapeutic development in obesity and its complications'.

Our key revisions include the following:

- i) Both the content and structure have undergone extensive edits. In regards to the oxyntomodulin physiology, while the mechanisms of action we discussed remained the same (appetite regulation, energy expenditure, and glucose homeostasis), we have provided a more oxyntomodulin-centric discussion. For example, we focused on the link between the physiology of oxyntomodulin and its link to NNMT, amino acids, and FGF21 in the energy expenditure section, which are biomarkers of glucagon receptor engagement, leading to its energy expenditure effect.
- ii) In light of the reviewers' feedback, we have expanded and clarified the molecular insights underlying oxyntomodulin, which we covered the development since its inception, to its processing and innate challenges which explain why oxyntomodulin has been lagging behind in its therapeutic translation compared to incretin-based agents.
- iii) Another key change we made was the therapeutic potential, we discussed the development of oxyntomodulin-based agents, leading up to the recent survodutide, emphasising on the ratio of the dual agonism and highlighting that the gastrointestinal adverse effects remain the key concern as we move forward. We also provided practical considerations on how this dual-agonist therapy can advance, potentially shaping obesity care, especially with the current focus on 'health gains' rather than solely on the weight loss effect.
- iv) We included two figures – i) physiology of oxyntomodulin and ii) timeline of oxyntomodulin-based from physiology to therapeutic development

Given the extent of these changes, we believe that a review of the untracked version of the manuscript will help improve the reviewing experience. We hope that our substantial changes addressed the concerns raised and that the editor and referees will find our revised work satisfactory.

Best regards
Martin TW Kueh
Carel W le Roux

Referee 1

Comment 1. There are a number of issues in this short overview on oxyntomodulin (OXM) that need to be addressed. There are no page numbers.

Response 1. Thank you for your review and comment. Page numbers have been added.

Comment 2. There are no figures. As for any review article, figures are critical to summarize and transport the main message(s) put forward by the authors.

Response 2. We have taken into consideration and summarised our messages in two figures (timeline of OXM discovery and therapy development and the physiological effects of OXM).

Comment 3. There are numerous previous review articles on the subject which provide good summaries on OXM biology and pharmacology, including PMID 23019069 (2012), 29412831 (2018), PMID 37917871 (2024)

Response 3. We agree with the reviewer. We have now made significant edits to our review, in terms of the content and structure, along with expansion and clarification into the molecular insights of OXM and the therapeutic translation and some practical considerations. We revised the discussion on how the mechanisms of action work. Focusing specifically on appetite regulation, we talked about the challenges in understanding this effect across different species, and then we touched on how recent evidence from bariatric surgery highlighted the potential significance of OXM in regulating appetite through gut-brain modulation. For energy expenditure, we emphasised that its effect primarily comes from GCGR, and how its associated biomarkers (NNMT, amino acids, and FGF21) are taking effect. For glucose homeostasis, we discussed the impact of OXM and what its effect is like when compared to GLP1 and glucagon.

Comment 4. The Introduction section is very brief and thus requires more detail about the molecular biology and molecular pharmacology of OXM. PMID 29412831 provides one summary "Oxyntomodulin is a product of the glucagon precursor, proglucagon, produced and released from the endocrine L-cells of the gut after enzymatic processing by the precursor prohormone convertase 1/3. It corresponds to the proglucagon sequence 33-69 and thus contains the entire glucagon sequence plus a C-terminal octapeptide, comprising in total 37 amino acids. As might have been expected, it has glucagon-like bioactivity, but also and more surprisingly also activates the receptor for GLP-1." The authors should also mention that OXM is one of 8 peptides proteolysed from the glucagon peptide (glicentin, glicentin-related polypeptide (GRPP), glucagon-like peptide 1 (GLP-1, incretin), glucagon-like peptide 1 (7-37), glucagon-like peptide 1 (7-36), glucagon-like peptide 2 (GLP-2) (PMID 10605628 and <https://www.uniprot.org/uniprotkb/P01275/entry>), and that proteolytic cleavage may be dependent of the cell type and organ, as in In contrast, in L-cells , proglucagon is cleaved by PC1/3 at Arg-Arg sites to yield glicentin, GRPP, oxyntomodulin (OXM), GLP-1, intervening peptide-2 (IP-2) and GLP-2 (PMID10499540, PMID 9074764). It would be helpful if the authors could summarize the molecular biology and molecular pharmacology in a figure.

Response 4. Thank you for the important suggestion. We have created a separate section to discuss molecular insights, with a focus on its development, processing and challenges (lines 81 to 140). These important points have been carefully evaluated and incorporated.

Comment 5. "...a randomised trial found significantly lower levels of OXM in glucose responders without significant changes in the levels of GLP-1 and glucagon.⁶⁷ The actions of GLP1 and glucagon on their respective receptors were insufficient to explain the decrease in blood glucose." This conclusion is overinterpreting the findings not what the study showed, the aim of which was "To investigate changes in plasma levels of PGDP in glycaemic responders versus non-responders." in prediabetic individuals with samples being collected for 2.5 hours after consuming a ketone β -hydroxybutyrate (KE β HB)-supplemented beverage. The authors concluded that "Oxyntomodulin is involved in lowering plasma glucose and may play an important role in diabetes remission" which also is not supported by data, because changes in circulating levels are affected by production and clearance and say nothing about peptide activity.

Response 5. We have made appropriate amendments, as shown in lines 290 to 303. We have removed these comments, and now focused on the mechanisms suggested by the authors (i.e., the potential central effect of OXM and CCK-OXM interaction), with additional expansion on these suggestions.

Comment 6. Lines 190ff and lines 212-220: These sections require more detail on the molecular biology and pharmacology of FGF-21, particularly the comprehensive literature published by Aimin Xu's group on the role of FGF-21 needs to be discussed here

Response 6. Thank you for the suggestion. We have cited literature from Amin Xu's group (lines 236 to 254), where we discussed the relevance of FGF21 in the energy expenditure effect of GCGR.

Comment 7. Lines 361 ff: "Conclusion: Emerging without a designated receptor and regarded as less potent to native ligands (i.e., GLP1 and glucagon). This statement contradicts the literature and the authors' own statements in the manuscript (Lines 60 ff: "60 Oxyntomodulin (OXM), produced by the endocrine cells in the gut, activates glucagon receptors (GCGR) and glucagon-like peptide-1 receptors (GLP1-R)." and Lines 75 ff: "Oxyntomodulin (OXM), a 37 amino-acid hormone derived from proglucagon splicing in intestinal endocrine cells, acts as an endogenous dual agonist on glucagon receptor (GCGR) and glucagon-like peptide-1 receptor (GLP1-R).1, 2 Its potency is lower than the native ligands, due to its C-terminal". Also see comment #4 above

Response 7. We have made amendments to the conclusion, as shown in lines 377 to 387, where we discussed the challenges faced and opportunities that arise, from the aspects of molecular to physiology and therapeutic.

Reviewer 2

We thank the reviewer for comprehensively commenting on our work, which has aided significantly in the quality, content and structure of the revised manuscript. Given the substantial nature of the changes, we decided to respond to the comments per the section.

The first part (comment 1 to 6) focused on the appetite regulation section of the review. Here, we discussed the initial discoveries and the challenging task of pinpointing the central effects, as well as some physiological differences across species. Lastly, we discussed that evidence from bariatric surgery suggests that OXM may have an important role in controlling appetite between the gut and the brain, although not causally linked.

Comment 1. L. 94: there is no evidence that oxyntomodulin actually binds to these regions of the brain

Response 1. We agree that it was an oversimplification stating the involvement of OXM in these regions. We have therefore revised where the agreed pathway is mediated through the arcuate nucleus (lines 155 to 156), and there were inconsistencies across species (lines 156 to 159).

Comment 2. L. 107-113 the authors apparently suppose that peripheral oxyntomodulin actually enters the arcuate nucleus, but there is no evidence for that (whereas oxyntomodulin may certainly interact with the GLP-1 receptors of the circumventricular organs)

Response 2. We have removed these sentences.

Comment 3. L. 114 "the interspecies expression of OXM in central ARC or peripheral total alimentary canal transit time reduction" I don't understand this sentence (and if I do, it is wrong)

Response 3. We have removed this sentence.

Comment 4. L. 120-130: the effects of OXM on the clock are demonstrated with pharmacological doses and do not reflect physiological actions of the peptide

Response 4. We have removed these sentences.

Comment 5. L. 132-143. It is a bit bold to ascribe the effects of bariatric surgery to the limited rises in low-affinity endogenous oxyntomodulin

Response 5. We agree with the reviewer. We presented evidence from bariatric surgery studies, described the results from these papers, and made cautious implications. We included a line suggesting that the interpretation should be approached with caution. (Line 168 to 185).

Comment 6. L. 146: the authors should note that glucagon interacts with both the glucagon and the GLP-1 receptors

Response 6. We have removed the mentioned sentence and have taken careful consideration throughout the paper to avoid any confusion.

The second part (comments 7 to 13) focused on the energy expenditure section of the review. We discussed the unique action of OXM by linking to the biomarker of GCGR engagement, namely NNMT, amino acids and FGF-21. We provided relevant physiological links and some key considerations (e.g., NNMT effect across MASLD spectrum, glucose & lipid-linked amino acids, FGF21 resistance).

Comment 7. I. 151-162: this section is very confusing and poorly written. It is impossible to understand the point (except that oxynto should be measured with specific sandwich assays - and such assays are currently not available (in a reliable form))

Response 7. We agree and the changes have been made. Now, the methodological limitations for OXM quantification are discussed from lines 115 to 120.

Comment 8. I. 164. One of the paradoxes in the literature was that the energy increasing activity of oxyntomodulin was found to be due to increased physical activity. This of course is difficult to explain and highly unlikely.

Response 8. We agree and hence, this explanation was not suggested to avoid confusion. We suggested that this might be due to the side effects, supported by data (lines 327 to 335).

Comment 9. I. 183-4: " Low blood insulin levels lead to an energy insufficiency, causing the release of glucagon to increase energy expenditure processes. This is a concentration-substantiated process mediated through brown adipose tissue or the sympathetic nervous system" I don't know what was the intention with this sentence, it is (again) incomprehensible, and at any rate not consistent with current physiologically knowledge

Response 9. These sentences have been removed.

Comment 10. I. 186: there is no evidence that glucagon is taken up by the liver cells (it acts on them)

Response 10. We have removed the mentioned sentence and have taken careful consideration throughout the paper to avoid any confusion.

Comment 11. I. 182-193: do the authors suggest that glucagon activates energy metabolism in humans via an action on the brown adipose tissue?

Response 11. We have removed the mentioned sentences and have taken careful consideration throughout the paper to avoid any confusion.

Comment 12. I. 198: do the authors really believe that glucagon acts via the FCXR? Glucagon increases fatty acid oxidation in the liver and inhibits de novo lipogenesis and VLDL export, via interaction with the GCGR and the cAMP pathways. Those provide good explanations for the effects

Response 12. We agree that the pathways do not link up and we have now removed the section on FXR. We emphasised 3 key intersections – NNMR, amino acids and FGF21, which have all been suggested as the biomarkers of energy-regulating biomarkers associated with GCGR agonistic engagement.

Comment 13. I. 235 "As discussed earlier, GLP1-R activation is rather insulin-centric, leading to glycogenesis and acetyl-CoA conversion to fatty acid-mediated through protein kinase C and PPAR gamma signalling". Again this is impossible to understand or wrong

Response 13. We have removed the edits accordingly and made amendments to improve the overall clarity of the physiological actions in regards to glucose homeostasis.

The third part (comment 14) focused on the glucose homeostasis section of the review. We discussed the discovery of OXM and its effect on glucose regulation. We also compared its effects with GLP1 and considered the potential central or peripheral links of OXM to induce glucose control effects, as well as the important considerations of glucagon effects.

Comment 14. I. 247: there is no evidence that GLP-1 acts directly on the liver

Response 14. We have removed the mentioned sentence and have taken careful consideration throughout the paper to avoid any confusion.

The fourth part (comment 15 to 18) focused on the therapeutic translation section of the review.

Comment 15. I320: endogenous oxyntomodulin is not substrate for DPP-4

Response 15. We agree and we have now removed the sentence.

Comment 16. I. 330: it should be noted that the development of cotadutide is currently put on hold

Response 16. We agree with the reviewer. Gastrointestinal side effects are a common challenge across these agents, including cotadutide, which we briefly discussed in lines 327 to 335. We believe

that this is something that needs to be carefully considered when moving forward with therapeutic decisions.

Comment 17. I. 346: "Overall, we are now witnessing the OXM-led physiological reprogramming and how its interplay with obesity associated pathology in human trials. " Another example of the problematic style of this review.

Response 17. We agree and we have now removed the sentence

Comment 18. Overall, Regarding the clinical application: it is rather simple: what happens depend of the ratio between glucagon activity and GLP-1 activity of the dual agonist; if there is too much glucagon all the problems with glycemic control, amino acid metabolism, side effects, and cardiac effects set in, but there is a stronger effects on hepatic lipid metabolism

Response 18. We discussed these important points in our new section on the progress of OXM-based therapies (lines 304 to 349). We started with the early studies shedding light on how OXM could be translated into therapy, followed by the struggles to identify a suitable glucagon-to-GLP1 ratio. We highlighted GI side effects as an important consideration, as well as effects related to cardiac, amino acid and lipid. Next, we highlighted on survodutide, which may be promising with the 1:8 (GCGR: GLP1R) ratio. Finally, we also provided some insights into the practical directions for the OXM-based approach (lines 351 to 375).

Dear Mr Kueh,

Re: JP-SR-2024-287407R1 "Oxyntomodulin physiology and its therapeutic development in obesity and its complications" by
Martin Tze Wah Kueh
Ming Chuen Chong
Alexander D Miras
Carel Le Roux

I am pleased to tell you that your Symposium Review article has been accepted for publication in The Journal of Physiology, subject to any modifications to the text that may be required by the Journal Office to conform to House rules.

NEW POLICY: In order to improve the transparency of its peer review process, The Journal of Physiology publishes online as supporting information the peer review history of all articles accepted for publication. Readers will have access to decision letters, including all Editors' comments and referee reports, for each version of the manuscript and any author responses to peer review comments. Referees can decide whether or not they wish to be named on the peer review history document.

The last Word version of the paper submitted will be used by the Production Editors to prepare your proof. When this is ready, you will receive an email containing a link to Wiley's Online Proofing System. The proof should be checked and corrected as quickly as possible.

All queries at proof stage should be sent to tjp@wiley.com.

The accepted version of the manuscript is the version that will be published online until the copy edited and typeset version is available. Authors should note that it is too late at this point to offer corrections prior to proofing. Major corrections at proof stage, such as changes to figures, will be referred to the Reviewing Editor for approval before they can be incorporated. Only minor changes, such as to style and consistency, should be made a proof stage. Changes that need to be made after proof stage will usually require a formal correction notice.

Are you on Twitter? Once your paper is online, why not share your achievement with your followers. Please tag The Journal (@jphysiol) in any tweets and we will share your accepted paper with our 30,000+ followers!

If you would like to receive our 'Research Roundup', a monthly newsletter highlighting the cutting-edge research published in The Physiological Society's family of journals (The Journal of Physiology, Experimental Physiology and Physiological Reports), please click this link, fill in your name and email address and select 'Research Roundup':
<https://www.physoc.org/journals-and-media/membernews/>

Yours sincerely,

Kim Barrett
Senior Editor
The Journal of Physiology

EDITOR COMMENTS

Thank you for your thoughtful response to the previous reviews. I anticipate that your article will be of substantial interest to our readership.

REFeree COMMENTS

Referee #1:

No further comments.

Referee #2:

No further comments.

* IMPORTANT NOTICE ABOUT OPEN ACCESS *

To assist authors whose funding agencies mandate public access to published research findings sooner than 12 months after publication, The Journal of Physiology allows authors to pay an open access (OA) fee to have their papers made freely available immediately on publication.

You will receive an email from Wiley with details on how to register or log-in to Wiley Authors Services where you will be able to place an OnlineOpen order.

You can check if your funder or institution has a Wiley Open Access Account here: <https://authorservices.wiley.com/author-resources/Journal-Authors/licensing-and-open-access/open-access/author-compliance-tool.html>.

Your article will be made Open Access upon publication, or as soon as payment is received.

If you wish to put your paper on an OA website such as PMC or UKPMC or your institutional repository within 12 months of publication you must pay the open access fee, which covers the cost of publication.

OnlineOpen articles are deposited in PubMed Central (PMC) and PMC mirror sites. Authors of OnlineOpen articles are permitted to post the final, published PDF of their article on a website, institutional repository, or other free public server, immediately on publication.

Note to NIH-funded authors: The Journal of Physiology is published on PMC 12 months after publication, NIH-funded authors DO NOT NEED to pay to publish and DO NOT NEED to post their accepted papers on PMC.

1st Confidential Review

23-Sep-2024